# Analysis of Data Reception in the Communication Layer Applied to an Architecture of Mobile Sensor Networks in Marine Environments

**DOI:** 10.3390/s23125480

**Published:** 2023-06-10

**Authors:** Abigail Elizabeth Pallares-Calvo, Blanca Esther Carvajal-Gámez, Octavio Gutiérrez-Frías, Dante Mujica-Vargas

**Affiliations:** 1Instituto Politécnico Nacional, Unidad Profesional Interdisciplinaria en Ingeniería y Tecnologías Avanzadas, Mexico City 07340, Mexico; becarvajal@ipn.mx (B.E.C.-G.); ogutierrezf@ipn.mx (O.G.-F.); 2TecNM-CENIDET, Interior Internado Palmira S/N, Col. Palmira, Cuernavaca 62490, Mexico; dante.mv@cenidet.tecnm.mx

**Keywords:** RFID, sensor networks, mobile sensor, static sensor, marine environments, data reception, monitoring

## Abstract

This paper is focused on the use of radio frequency identification (RFID) technology operating at 125 kHz in a communication layer for a network of mobile and static nodes in marine environments, with a specific focus on the Underwater Internet of Things (UIoT). The analysis is divided into two main sections: characterizing the penetration depth at different frequencies and evaluating the probabilities of data reception between antennas of static nodes and a terrestrial antenna considering the line of sight (LoS) between antennas. The results indicate that the use of RFID technology at 125 kHz allows for data reception with a penetration depth of 0.6116 dB/m, demonstrating its suitability for data communication in marine environments. In the second part of the analysis, we examine the probabilities of data reception between static-node antennas at different heights and a terrestrial antenna at a specific height. Wave samples recorded in Playa Sisal, Yucatan, Mexico, are used for this analysis. The findings show a maximum reception probability of 94.5% between static nodes with an antenna at a height of 0 m and a 100% data reception probability between a static node and the terrestrial antenna when the static-node antennas are optimally positioned at a height of 1 m above sea level. Overall, this paper provides valuable insights into the application of RFID technology in marine environments for the UIoT, considering the minimization of impacts on marine fauna. The results suggest that by adjusting the characteristics of the RFID system, the proposed architecture can be effectively implemented to expand the monitoring area, considering variables both underwater and on the surface of the marine environment.

## 1. Introduction

We are surrounded by sensors that allow us to collect data in specific areas [1]; these sensors are connected in networks to interact with other devices without human intervention, which is known as the Internet of Things (IoT) [2]. Sensors that can communicate with other devices without human intervention are called smart sensors [3] and are used in different areas, such as health, industry, engineering, and biology, to name a few. An example of a biological application is presented in [4], in which sensors were employed to communicate with servers within a five-layer IoT architecture (sensing, communication, network, storage, and application), which was used for sheep monitoring. Sensors collected data on the location, posture, and behavior of the sheep with frequencies in the order of MHz and transmitted the data to be processed, saved, and viewed by the end user.

As in terrestrial environments, where smart sensors interact without human intervention, the same principle is applied in aquatic environments to determine the characteristics of the environment and monitor animals, people, plants, and objects in rivers and/or oceans, all using the paradigm known as the UIoT [5]. The implementation of the UIoT paradigm can be beneficial in the context of autonomous underwater vehicles (AUVs) for data management in which provide a user interface for managing and organizing this data [6] or in algorithms for data collection to improve data collection efficiency and overcome the limitations caused by node mobility [7]. An example of an implementation of intelligent sensor communication in marine environments is provided in [8], in which ocean water quality was monitored through a two-layer architecture (sensing and communication). Sensors collected temperature, dissolved oxygen, pH, and turbidity data from the water, which were sent to the cloud via Wi-Fi communication. Another example is presented in [9], in which a five-layer architecture (sensing layer, communication layer, networking layer, fusion layer, and application layer) to record temperature, pH, and other types of data from sensors, which were transmitted to servers through underwater acoustic and/or optical communication for storage and analysis. Another example is presented in [10], in which the authors analyzed the impact of waves on communication between a buoy and an antenna in cellular communication at MHz and GHz frequencies. Effects of wave interference on the quality and reliability of the communication link were observed. However, it is worth noting that implementing higher frequencies in aquatic environments can have negative consequences, including temporary damage to the auditory system or permanent damage to the nervous or auditory tissue of animals, as well as disorientation of migrating animals [11,12]. Therefore, it is essential to consider the potential environmental impact and harm to wildlife when deploying communication systems in aquatic environments. In [13], an RFID communication system employed in saltwater was analyzed using frequencies of 134.5 kHz and 13.56 MHz. The study utilized two RFID readers, MRD2EVM operating at 134.2 kHz and Pepper Wireless C1 USB operating at 13.56 MHz. The readers were submerged in different types of water, and measurements of magnetic fields were taken at distances ranging from 0 to 10 cm. In [14], marine corrosion was monitored using an RFID system. The study involved submerging RFID tag 28340, RFID tag mifare1 S50, and RFID reader/writer MFRC 522 in both fresh and saltwater. The aim was to assess the reading range of the RFID systems in each of the media. The results showed that long reading ranges were achieved in both fresh and saltwater at low frequencies. The authors of [15] proposed a wireless communication system based on low-power magnetic induction for the transmission and retrieval of data from autonomous underwater vehicles (AUVs). The study involved mathematically characterizing the dynamics of the MI channel and deriving the available bandwidth based on the coupling coefficient between two coil antennas. The authors also developed a software-defined MI communication test bed system using MATLAB and USRP and verified its performance through simulations. The simulations conducted in the study yielded a transmission simulation distance of 0.81 m.

However, the works discussed above implemented signals with frequencies in the MHz or GHz range, which can have adverse effects on marine fauna. On the other hand, studies implementing signals in the kHz range are limited by the capabilities of RFID readers and magnetic induction design and focus solely on data reception underwater without considering subsequent data transmission to the Earth’s surface. To address these limitations, in this work, we propose a four-layer architecture comprising detection, communication, networking, and storage. Our focus is on characterizing the communication of smart sensors, considering two types of data reception. The first type involves the use of RFID technology with magnetic induction at a frequency of 125 kHz, enabling data reception in a saline marine environment between a mobile node and a static node at a known geographical location. We analyze the penetration depth of magnetic induction signals to determine the range of data reception between these nodes. The second type of data reception utilizes Wi-Fi technology to facilitate communication between static nodes and an antenna located on the Earth’s surface. The impact of ocean waves on data reception is analyzed using wave data recorded by the Laboratory of Coastal Engineering and Processes located at Sisal Beach, Yucatan, Mexico. Based on these data, the probability of data reception via line of sight between the antennas is calculated.

The research aims to validate the reception of data in a sensor network architecture in two distinct environments: a marine environment and an above-sea environment. In the marine environment, the data signals operate at frequencies in the order of kHz, specifically chosen to minimize any potential impact on marine fauna. On the other hand, in the above-sea environment, the data reception is affected by the presence of waves, which can potentially disrupt the reception of data signals. This contributes to the advancement of sensor network technology for marine and surface applications.

The contributions of this work are summarized as follows:Validation of a sensor network architecture to be implemented in marine environments;Validation of low-frequency data reception, which is useful in marine environments;Verification of the viability of the two proposed types of communication in the smart sensor architecture.

By addressing these aspects, this research offers valuable insights into the design and implementation of a sensor network architecture for marine environments. The results presented herein validate the reception of data at low frequencies as a useful approach in marine environments and confirm the viability of the proposed communication methods for smart sensors in such environments.

The remainder of this paper is organized as follows. In Section 2, the design of the proposed smart sensor architecture is described. In Section 3, simulations of the communication between mobile nodes and the static node are described, and the penetration depth and LoS of the communication between the static nodes and the terrestrial antenna are analyzed, considering the environmental conditions. In Section 4, we present and discuss the results. Finally, conclusions are drawn in Section 5.

## 2. Materials and Methods

In this section, we present the architecture of the proposed sensor network, which, by implementing magnetic induction technology at a frequency of 125 kHz, does not affect marine fauna [16]; therefore, we refer to the sensors as smart friendly sensors. The proposed smart sensors are intended to monitor objects through nodes that are in motion and send the collected data to nodes located at specific points, which, in turn, forward the data to a storage cloud. Subsequently, we present the characterization of the sensor network. Firstly, the parameters required for the reception of data in aquatic environments between the fixed and mobile nodes are presented. Secondly, the relationship of waves with LoS is established based on real results from Sisal Beach, Yucatán, Mexico, to determine the probability of reception between fixed nodes and between fixed nodes and a terrestrial antenna.

### 2.1. Architecture of the Sensor Network

The architecture of the proposed sensor network involves the collection of data from an object of interest through the implementation of a network of nodes that use RFID technology at a frequency of 125 kHz, which is commonly used to monitor species in marine areas for three reasons. First, the proposed approach results in a reduction in the reading range of 30 to 40% [17]. Secondly, according to the ISO (International Organization for Standardization standard) standards ISO 11784 and ISO 11785, this frequency range is implemented in the monitoring of animals [16]. Lastly, this frequency range does not cause temporary or permanent damage to the nervous systems, ear canals, and/or organs of marine fauna [11].

The proposed architecture is depicted in Figure 1, showcasing a scenario in which moving sensors (symbolized by fish) equipped with mobile nodes collect data. The collected data are transmitted to buoys (static nodes) positioned in specific locations. Communication between the mobile nodes and static nodes is facilitated through RFID technology, as represented by the green dotted line indicating data reception. When the static nodes receive data, they forward them to other static nodes using Wi-Fi connectivity. This data transmission among static nodes is illustrated by the purple dotted lines in the figure. The data eventually reach a terrestrial antenna, where they are stored for further analysis.

Figure 2 shows the architecture the proposed UIoT system, which consists of four layers: detection, communication, networking, and cloud storage. The detection layer is responsible for collecting data from various types of sensors that detect objects, animals, people, or the environment. These sensors are implemented through nodes that are in motion (mobile nodes) and placed on the objects of interest. These mobile nodes interact with nodes that are fixed to known geographic points (static nodes). The communication layer is a hybrid communication system that operates via magnetic induction between the mobile nodes and the static node and via Wi-Fi modules in the communication between the static nodes and the terrestrial antenna. This layer facilitates the transfer of data collected by the sensors between the nodes and the cloud storage layer. The networking layer is composed of a set of nodes that provide connections between the defined points. This layer is responsible for managing the network topology, routing of data, and maintenance of the network. Finally, the cloud storage layer is responsible for receiving the data sent by the nodes and storing them in the cloud. The stored data can then be analyzed for later use. This layer enables the user to remotely access the data and perform various analysis tasks. Overall, the proposed architecture represents a comprehensive system that integrates various layers to facilitate the collection, communication, storage, and analysis of data in marine environments using RFID technology.

### 2.2. Characterization of the Data Reception of the Communication Layer for a Sensor Network for Aquatic Environments

To validate the successful reception of data in the proposed architecture for the two different environments, first, a simulation of the penetration depth at different frequencies is performed, considering the parameters that can affect the transmission by magnetic induction in marine environments, this simulation makes it possible to verify whether the frequency that is proposed can be implemented in the reception of data between the nodes in a marine environment. Secondly, an analysis is carried out for the reception of data between the nodes and the terrestrial antenna, in which the real waves that may exist and affect the LoS between the devices are considered, and whether the heights of the antennas of the nodes generate changes in data reception.

#### 2.2.1. Characterization of Data Reception between the Mobile Nodes and the Static Node

The reception of data between the mobile nodes and the static node in the proposed architecture is facilitated by RFID technology, which relies on magnetic induction. Magnetic induction can be influenced by the characteristics of the medium through which the signal travels. For instance, seawater, which has a high salt content, exhibits an average conductivity of 4 Siemens per meter (S/m). In contrast, freshwater typically has a conductivity of around 0.01 S/m. These conductivity differences can impact the performance and range of RFID communication in marine environments [18].

To understand how signals propagate in water, the propagation constant can be determined [19]:(1)$$\gamma =\sqrt{j\omega \mu \left(\sigma +j\omega \varepsilon \right)}=\alpha +j\beta \left(m^{-1}\right)$$
where *σ* is the conductivity of the water in S/m, *μ* is the permeability in N/A^2^, and *ε* is the permittivity in F/m. If the medium is not free space, the propagation constant (*σ*) is a complex quantity, with *α* (the attenuation factor or propagation loss) and *β* (the phase factor) defined by Equations (2) and (3), respectively [20]:(2)$$\alpha =\omega \sqrt{\mu \varepsilon }{\left[\frac{1}{2}\left(\sqrt{1+{\left(\frac{\sigma }{\omega \varepsilon }\right)}^{2}}\right)-1\right]}^{\frac{1}{2}}\left(\frac{\mathrm{dB}}{m}\right)\;$$
(3)$$\beta =\omega \sqrt{\mu \varepsilon }{\left[\frac{1}{2}\left(\sqrt{1+{\left(\frac{\sigma }{\omega \varepsilon }\right)}^{2}}\right)+1\right]}^{\frac{1}{2}}\left(\frac{\mathrm{rad}}{m}\right)$$

In [17], the attenuation of RFID in saltwater is described without separating it into two factors, as shown below:(4)$$\alpha =0.0173\sqrt{f\times \sigma }\;\frac{\mathrm{dB}}{m}$$
where *σ* is the conductivity of the water in S/m, and f is the frequency. Equations (2) and (4) in the referenced work define propagation loss by considering various variables.

To determine the most suitable characterization for underwater data reception in the proposed architecture, Algorithm 1 is implemented. This algorithm involves comparing a range of frequencies and evaluating which equations proposed in the literature best represent the data obtained in marine environments.

By conducting this comparison, the algorithm helps to identify the equation or model that provides the closest match to the observed data and accurately represents the characteristics of underwater data reception in the proposed architecture. This process ensures that the chosen characterization aligns with the specific requirements and conditions of the marine environment.
**Algorithm 1.** Algorithm to determine the depth range for data reception in a saltwater environment.**Data:**f0 sample frequenciesσ medium conductivityμ medium permeabilityε permittivity of the mediumateσ penetration depth considering σate penetration depth considering σ, μ and ε**begin**
For the penetration depth considering σ  **for** k = 0:1: f0final (f0final maximum value of f0) **do**   ateσ=0.0173f0·σ  **end for**For the penetration depth considering σ,μ0 and ε0  **for** k = 0:1: f0final
**do**   ω=2·π·f0   ate=ωμ·ε121+σω·ε2−11/2  **end for****end**

#### 2.2.2. Characterization of Data Reception between Static Nodes and the Terrestrial Antenna

In the analysis of data reception at the sea surface, the focus is on the communication between static nodes or between a static node and a ground antenna. A specific scenario is considered in which the data reception between a buoy and an antenna is used as a reference. This scenario considers the presence of sea waves between the buoy and the antenna, which affects LoS communication.

Figure 3 illustrates the considered scenario, highlighting the interaction of the sea waves with the communication link between the buoy and the antenna. The position of the buoy relative to the waves is crucial for the quality of the reception, as shown in Figure 3; the blue dotted line represents the blocking of the LoS, indicating when the wave obstructs the direct communication path. The alpha and beta angles are used to describe the geometric relationship between the terrestrial antenna, the sea surface, and the line of sight. When the buoy is positioned on the crest of a wave, the reception is improved due to the elevated position. Conversely, when the buoy is in the trough of a wave, reception can be hindered or blocked due to being at a lower position. Both scenarios, i.e., the buoy either on the crest or in the trough of a wave, need to be analyzed to understand their impact on data reception. By considering these factors, researchers can assess the reliability and performance of the communication link at the surface of the water.

An analysis was carried out considering the real mechanism of the fluid dynamics of the ocean waves and the variation in the elevation of the surface in a highly dynamic oceanic environment. However, due to the types of variables and the movement of the fluid (saltwater), a method was used to achieve a precise study without delving into fluid dynamics. For this reason, precisely how the seawater surface elevation varies with time was investigated through statistical models. To this end, an analysis of wave spectra was implemented, for which the spectra of two parameters were used, the Bretschneider spectrum, also known as the ISSC spectrum (represented by the significant wave height (SWH) and the average period) [21], which represents wave conditions in the open sea, as expressed by
(5)$$S_{\eta }\left(\omega \right)=\frac{5}{16}\frac{H_{s}^{2}\omega_{p}^{4}}{\omega^{5}}exp\left(-\frac{5}{4}{\left(\frac{\omega_{p}}{\omega }\right)}^{4}\right)\;\mathrm{in}\;\frac{m^{2}}{\left(\frac{\mathrm{rad}}{s}\right)}$$
where $H_{s}$ is the SWH in meters, also known as $H_{1/3}$, which is traditionally defined as the mean wave height (trough to crest) of the lowest third wave height, and $\omega_{p}$ is the modal (peak) angular frequency in rad/s. The peak period is expressed as $T_{p}=2\pi /\omega_{p}$.

For a regular monochromatic ocean wave of amplitude A, angular frequency *ω*, and phase constant *ε*, the moving ocean surface elevation (*η*(*t*)) can be described as
(6)$$\eta \left(x,y,t\right)=R\left\{A\;exp\left(j*\left(-kxcos\theta -kysin\theta +\omega t+\varepsilon \right)\right)\right\}$$
where θ is the direction of wave propagation from the *x* axis. For a case in which ε=0, the directionality is not considered when the propagation direction of the ocean is normal to the coast [22]; the variation of the water surface at the origin (also the location of the buoy) when x=0 is expressed as
(7)$$\eta \left(0,t\right)=A\;cos\left(\omega t\right)$$

A real ocean wave is made up of a large number of frequency components ($N_{f}$). Therefore, the ocean wave is the sum of the ocean waves of all frequency components and is expressed as in [23]:(8)$$\eta \left(x,t\right)=\overset{N_{f}}{\underset{i=1}{\sum }}a_{i}\;cos\left(2\pi f_{i}t+k_{i}x+\alpha_{i}\right)$$
where $a_{i}$ is the amplitude, $k_{i}=2\pi /\lambda_{i}$ is the wavenumber, and $\alpha_{i}$ is the phase of the ith frequency component ($f_{i}$). Likewise, the expected amplitude ($\mu_{i}$) of each frequency component can be calculated as in [22]:(9)$$\mu_{i}=\sqrt{2\cdot S_{\eta }\left(\omega_{i}\right)\cdot \Delta \omega }$$
where $S_{\eta }\left(\omega_{i}\right)$ is the spectrum of ocean waves that can be obtained using (5), and Δω is the width of the frequency range of the spectrum ($S_{\eta }\left(\omega \right)$). At the location of the buoy, x=0; therefore,
(10)$$\eta \left(0,t\right)=\overset{N_{f}}{\underset{i=1}{\sum }}a_{i}\;cos\left(\omega_{i}+\alpha_{i}\right)$$

At any other location, such as $x_{n}$,
(11)$$\eta \left(x_{n},t\right)=\overset{N_{f}}{\underset{i=1}{\sum }}a_{i}\;cos\left(k_{i}x_{n}+\alpha_{i}\right)$$

Consequently, once the combination of ($H_{s}$, $T_{p}$) is defined using (9)–(11), a time-varying ocean wave can be simulated at any location between the buoy and the antenna in the time domain, with the adjustable time interval variable as adjustable resolution.

As shown in Figure 3, when the reception of data in the LoS is deactivated by an ocean wave blocker that appears at $x_{n}$, β>α, where β is the angle between the LoS and sea level, and *α* is the angle of the line connecting the sea surface at $x=x_{n}$ and the cell tower antenna to sea level. This relationship leads to the following wave-blocking criterion:(12)$$\frac{h_{twr}-\left(\eta \left(0,t\right)+h_{a}\right)}{d}>\frac{h_{twr}-\eta \left(x_{n},t\right)}{d-x_{n}}$$
where $h_{a}$ is the effective height of the buoy antenna and is measured vertically from sea level; $h_{twr}$ is the height of the cell tower; d is the distance between the buoy and the cell tower; $x_{n}$ is the horizontal location of the ocean wave-blocking point; and $\eta \left(0,\;t\right)$ and $\eta \left(x_{n},\;t\right)$ are the ocean surface elevation at the location of the buoy and the ocean wave blocker, respectively.

The equations presented in this work are used to sample ocean waves and determine if there is an LoS and data reception at the buoy location through the implementation of Algorithm 2, which utilizes a database of wave data specific to the location.

The purpose of the algorithm is to substitute the database values and consider the distances between antennas and the height differences between them to identify areas where data reception is blocked. Equation (12) is used to establish the relationship between these parameters.

Once the locations with no data reception blocking have been identified, the Rayleigh probability is calculated. This probability determines the likelihood of data reception between the static nodes and the antenna through LoS conditions.

The Rayleigh probability is commonly used in scenarios involving antennas due to the specific conditions and characteristics associated with wireless communication, providing a statistical model that can accurately represent the probability of data reception in such scenarios.
**Algorithm 2.** Algorithm to determine the occurrence of data reception between the static nodes and the antenna.**Data:**Hs significant wave heightTp peak periodhtwr terrestrial antenna heightha node antenna heightd distance between antenna and nodeNrp number of random realizations**begin** Generate wave number ki for each frequency component ωi set ki:=ω i2g; Na :=0; Nf:=number of frequency components  **while**
Na<Nrp
**do**   **for**
t=0:Δt:T    μi:=2·Sηωi·Δω    ai:=raylrndμi    αi:=rand1,Nf·2·π    η0,t=∑i=1Nfaicosωi+kix+αi    ηxn,t=∑i=1Nfaicosωi+kixn+αiFind the blocker at the current time instance    **for**
xn=1:d
**do**     **if**
htwr−η0,t+had>htwr−ηxn,td−xn
**then**  The distance and height of the blocker landing are saved to be subtracted from the component number.     **end if**    **end for**   **end for**  Na:=Na+1  Regenerate a set of realizations of ai and αi random  **end while****end**

## 3. Results

MATLAB^®^ software version 2022a was run on a Lenovo Ideapad Gaming 3 computer with an AMD Ryzen 5 processor 4000 series for different test scenarios, as described below.

In the first scenario, data reception between mobile and static nodes was analyzed. For this scenario, an RFID reader with an omnidirectional antenna placed at a known geographical point is considered; the RFID reader is submerged in water, while the other node is moving in the water, with another omnidirectional antenna containing the label (see Figure 4; the green and orange dotted lines represent the reception of data between the mobile and static nodes). Analysis was carried out to determine whether the particles of the aquatic environment at the proposed frequency allow for the passage of a signal. This scenario was implemented using Algorithm 1, in which the penetration depth of frequencies ranging from 0 to 10 GHz is obtained, considering the characteristics of the environment with average conductivity values in the sea of 4 S/m, the relative permittivity of 81, and relative permeability of 0.999991 [24].

The depth of penetration is represented in Figure 5 based on the implementation of Algorithm 1. The blue curve represents the depth of penetration when only the conductivity of the medium is considered, whereas the red curve represents the depth of penetration, taking into account the conductivity, permittivity, and permeability of the medium. For a frequency of 8.00798 GHz, considering only the conductivity, the penetration depth is 154.813 dB⁄m. However, when all three parameters (conductivity, permittivity, and permeability) are considered, the penetration depth is slightly reduced to 151.272 dB⁄m, suggesting that the additional factors of permittivity and permeability have a slight impact on the depth of penetration.

Figure 6 shows the behavior of the penetration depth with a 125 kHz signal of interest; the blue line indicates that the penetration depth reaches a value of 0.6116 dB/m, whereas the red line shows a value of 0+1.5693i dB/m. This comparison shows an abrupt change when incorporating conductivity, permittivity, and permeability compared to considering only conductivity.

In the second scenario, the focus is on the reception of data between the static nodes and the terrestrial antenna, considering the presence of waves in the medium. To determine the feasibility of reception despite the waves, the probability of LoS communication is considered. The analysis of this test scenario is divided into two parts.

The first half of the scenario involves the reception of data between the static nodes, as illustrated in Figure 7. The purple dotted line represents the data reception between the antennas, and “dn” denotes the distance between the static nodes. Because commercial RFID readers typically have a range of 1 m in terrestrial environments, and considering a penetration depth of 0.6116 dB/m for data reception between the static node and the mobile nodes, we recommend that the static nodes be positioned 1 m apart.

To determine the ideal antenna height for optimal data reception, the antenna of one static node is set at 0 m above sea level, and the antenna of the other static node is tested at three possible heights relative to sea level: 0 m, 0.5 m, and 1 m. The objective is to find the antenna height that offers the best data reception performance in the given scenario.

The simulation uses ocean wave data collected from the Gulf of Mexico region, specifically Sisal Beach on the Yucatan peninsula. The data were obtained from the Southeast Coastal Observatory [25], covering the period from March to November 2019. Figure 8 depicts the geographical location of Sisal Beach (indicated in red), and Table 1 displays a fragment of the database.

For this test scenario, a sample of 1000 SWH values recorded at Sisal Beach, Yucatan, Mexico, was implemented via Equation (5) using MATLAB 20222a software, yielding the data presented in Figure 9, with ocean waves at a distance of one meter, where the distance of x represents the offset angle added to the current wave by the previous wave, providing a true representation of how the waves affect the LoS for data reception between the antennas of the static nodes that are located on the sea surface. In Figure 9, 0 represents sea level; the largest value was at 0.2455 m, which represents the highest peak in the sample, and −0.2214 m is the lowest value in the spectrum, which represents the deepest valley of the sample. This simulation provides valuable insights into the characteristics of the ocean waves at Sisal Beach, enabling a better understanding of how these waves influence the LoS for data reception between the antennas of the static nodes. These data contribute to the assessment of the underwater communication environment in the proposed architecture by providing a realistic representation of wave behavior.

Once the wave sample has been obtained, the probability of data reception between the static nodes is obtained by implementing the Rayleigh distribution. For this test scenario, an antenna is proposed at three different heights—0 m, 0.5 m, and 1 m—all with a separation between nodes of 1 m. Using Algorithm 2 in MATLAB, the number of times the LoS is not blocked is obtained, and the probability density function (PDF) of the data reception is calculated. Figure 10 illustrates the impact of difference between antenna heights on data reception, with the blue line representing a node antenna height of 1 m, the red line representing a height of 0.5 m, and the yellow line representing a height of 0 m.

The corresponding probabilities of data reception for these antenna heights are summarized in Table 2, which presents the average results from 10 rounds of experiments. For an antenna height of 0 m, the probability of data reception is 54.2%, with the occurrence range centered around a height difference of 0 to 0.2886 m. At an antenna height of 0.5 m, the probability increases to 70.6%, centered on a height difference of 0 to 0.2573 m. Finally, at an antenna height of 1 m, the probability reaches 94.5%, with a range of height difference between antennas ranging from 0 to 0.2362 m. Comparing these probabilities reveals that higher antenna heights result in steeper slopes of the plots, indicating improved data reception as the antenna height increases. This suggests that raising the antenna height can significantly enhance the probability of data reception.

Part two of the second test scenario focuses on the data reception between the static node and the ground antenna, and as in the first half, the static node device is considered to be at sea level, the height of the ground antenna device between is considered to be between zero and one meter (i.e., sea level), and the terrestrial antenna is considered to be at a height of 45 m above sea level. Figure 11 shows a depiction of the considered scenario, where d is the distance between the terrestrial antenna and the static node, Xn is the height of the wave above sea level, the continuous blue line is the LoS without blocking, the dotted blue line is the blocked LoS, alpha is the angle of the line between the sea surface and the terrestrial antenna, and beta is the angle between the LoS and the horizontal sea level.

For this scenario, Algorithm 2 is implemented with the data reception between the static node and the antenna considered as parameters. The wave sample for this scenario is 1 km, as shown in Figure 12, where x represents the phase angle added to the current wave by the previous wave, providing a real representation of the waves that affect the LoS for data reception between the static node and the terrestrial antenna; likewise, 0 represents sea level, with a maximum value of 0.3763 m and a minimum value of −0.3735 m.

The probability of data reception between the static node and the terrestrial antenna is calculated using the sample wave. The conditions include a terrestrial antenna height of 45 m, and we consider three different heights for the node antenna: 0 m, 0.5 m, and 1 m. By implementing Algorithm 2 in MATLAB, we obtain the probability distributions, which are depicted in Figure 13, and which illustrate the impact of differences in antenna height on data reception between the node and terrestrial antenna, with the yellow line representing a node antenna height of 0 m, the red line representing a height of 0.5 m, and the blue line representing a height of 1 m. The corresponding probabilities of data reception for these antenna heights are summarized in Table 3, which presents the average results from 10 rounds of experiments. For an antenna height of 0 m, the probability of data reception is 98.1%. At an antenna height of 0.5 m, the probability increases to 99.2%. Finally, at an antenna height of 1 m, the probability reaches 100%. This suggests that raising the antenna height can significantly enhance the probability of data reception.

## 4. Discussion

The characterization of penetration depth at different frequencies reveals important insights into the behavior of signals in marine environments. At high frequencies such as $9.01658\times 10^{9}$ GHz when considering the values of *µ*, *ε*, and *σ*, the penetration depth is determined to be 162.195 dB/m. If only the value of *σ* is considered at this frequency, the penetration depth is slightly higher, at 164.273 dB/m.

Similarly, at a frequency of $8.00798\times 10^{9}$ GHz, considering *µ*, *ε*, and *σ*, the penetration depth is measured to be 151.272 dB/m. When only considering the value of *σ*, the penetration depth is slightly higher, at 154.813 dB/m.

At a frequency of 125 kHz, the penetration depth in the marine environment is expressed as 0+1.5693i dB/m, where the imaginary component represents a complex value. According to the literature [26], complex values tend toward infinity, indicating zero penetration depth. However, some studies [27] have demonstrated signal penetration in the marine environment considering only *σ*, resulting in a penetration depth of 0.6116 dB/m. Table 4 provides a comparison with other relevant works, analyzing a real database and exploring data reception both underwater and on the surface without harming the fauna in the environment.

In the analysis of data reception between nodes, the height of the antenna plays a crucial role. When considering three different heights, a height of 0 m, there is a 54.2% probability of data reception. As the antenna height increases, the probability of reception also increases, reaching a reception probability of 94.5% at higher antenna heights.

This probability trend is also reflected in the reception of data between the terrestrial antenna and the static node. At a height of 0 m, the probability of reception is 98.1%, indicating a high likelihood of successful data reception. However, when the antenna height is increased to 1 m, the probability of reception reaches 100%, ensuring a guaranteed reception of data between the terrestrial antenna and the static node.

These probabilities highlight the importance of antenna height in achieving reliable and consistent data reception between nodes in the given scenario.

Likewise, the use of ultrasonic signals for communication in marine environments is not considered viable due to the potential harm it can cause to marine fauna. Ultrasonic signals can lead to temporary hearing loss, organ damage, and disorientation in cetaceans [11,12]. Additionally, ultrasonic communication is susceptible to various factors, such as multipath effects, Doppler shift, temperature, pressure, salinity, and environmental noise [28]. Therefore, any analysis or design involving communication in marine environments must carefully consider all these variables to ensure the safety and well-being of marine fauna.

Table 4 provides a comparison of the results obtained in the current investigation with the results of previous studies in the literature. The parameters taken into consideration for the comparison include the propagation method, transmission frequency, penetration depth, underwater data reception, data reception on the sea surface, and the availability of a public database wave sample. The table shows that all studies, except for [8], employ magnetic induction for signal propagation. Additionally, only [12] operates at a frequency of 125 kHz. The investigations in [8,12] consider the depth of penetration, but the current investigation achieves a greater signal penetration. Furthermore, the current investigation is the only one that accounts for data reception in two different environments and implements a public database of a real beach for conducting simulations.

This suggests that the current investigation stands out in several aspects compared to the previous studies listed in the table. It utilizes a different propagation method, operates at a frequency that does not affect marine fauna, achieves greater penetration depth, incorporates data reception in multiple environments, and utilizes a publicly available database for realistic beach simulations. These distinctions highlight the novelty and potential contributions of the current investigation.

## 5. Conclusions

This paper discussed data reception in a sensor network architecture for marine environments. It explored two different scenarios: data reception using magnetic induction in an underwater environment, and data reception between nodes in the presence of beach waves.

In the first scenario, the reception of data using magnetic induction was examined at different frequencies in an underwater environment. It was determined that magnetic induction is primarily affected by conductivity, and the penetration depths of signals at 125 kHz and 8 GHz were compared. The analysis revealed that the 125 kHz signal had a penetration depth of 0.6116 dB/m, which was lower than that of the 8 GHz signal, which had a penetration depth of 154.813 dB/m. However, it was highlighted that the 125 kHz signal did not affect marine fauna as much as signals in the GHz range.

In the second scenario, data reception between nodes was investigated considering the waves at Sisal beach, Yucatan, Mexico. A sample of 1000 waves was collected to understand their impact on LoS data reception. Three different heights were proposed for the node antennas and analyzed the reception probabilities. The results indicate that at 0 m antenna height, the reception probability was 54.2%, while at 0.5 m, the probability increased to 70.6%, and at 1 m, it further improved to 94.5%. This demonstrates that higher antenna heights result in better data reception.

Furthermore, the paper examined data reception between a node and a terrestrial antenna. The same conditions as those between the nodes were maintained but a distance of 1 km was considered between the objects and a height of 45 m was considered for the terrestrial antenna. The analysis showed that at a node antenna height of 0 m, the reception probability was 98.1%, at 0.5 m, it increased to 99.2%, and at 1 m, it reached 100%. This highlights that higher antenna heights lead to improved data reception in this scenario as well.

This research provides valuable insights and can serve as a guideline for implementing the proposed architecture, considering the parameters studied. Future work will involve validating the data frame, considering additional variables that may affect data reception, and expanding the sample of waves beyond 1000.

Finally, the results are limited to the considerations proposed, such as the use of RFID technology for commercial purposes, the specific beach wave database used, and the proposed node antenna heights. However, the analysis can be reproduced by altering these factors, such as using a different beach database, considering RFID readers with a range exceeding one meter, or exploring alternative node antenna heights to achieve an extended range.

## Figures and Tables

**Figure 1 sensors-23-05480-f001:**
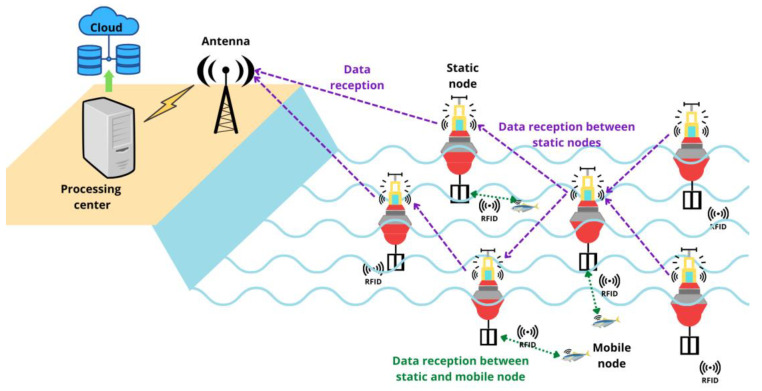
Architecture and implementation of the model.

**Figure 2 sensors-23-05480-f002:**
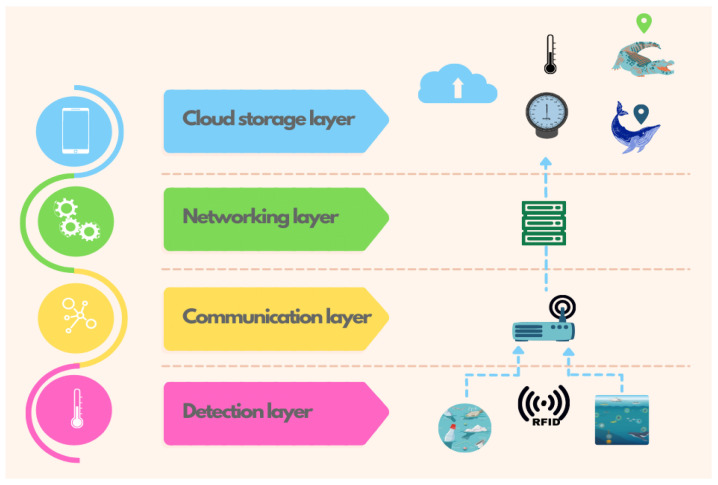
Layers of the proposed architecture.

**Figure 3 sensors-23-05480-f003:**
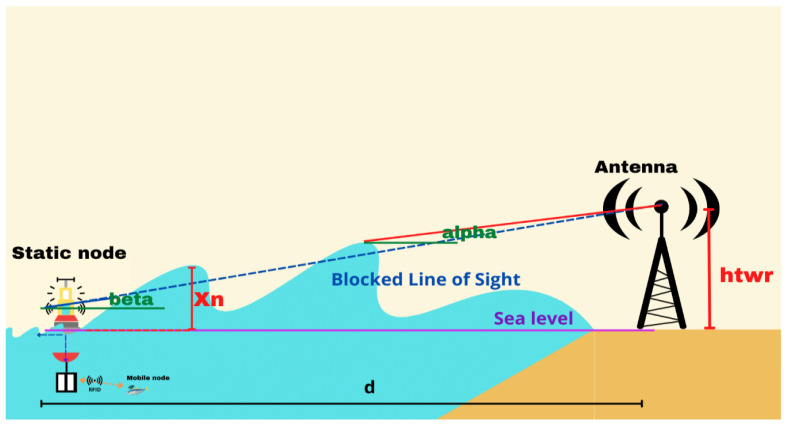
Communication between the buoy and the antenna considered in the analysis.

**Figure 4 sensors-23-05480-f004:**
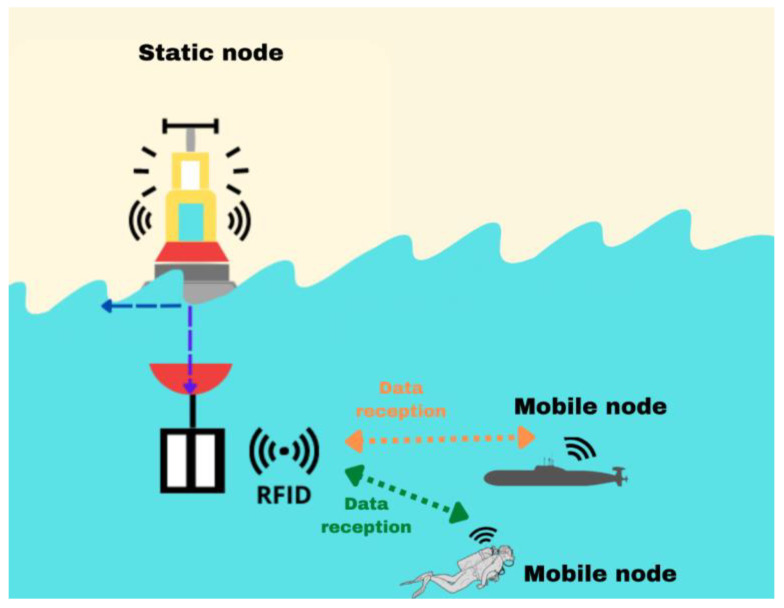
Description of the experiments.

**Figure 5 sensors-23-05480-f005:**
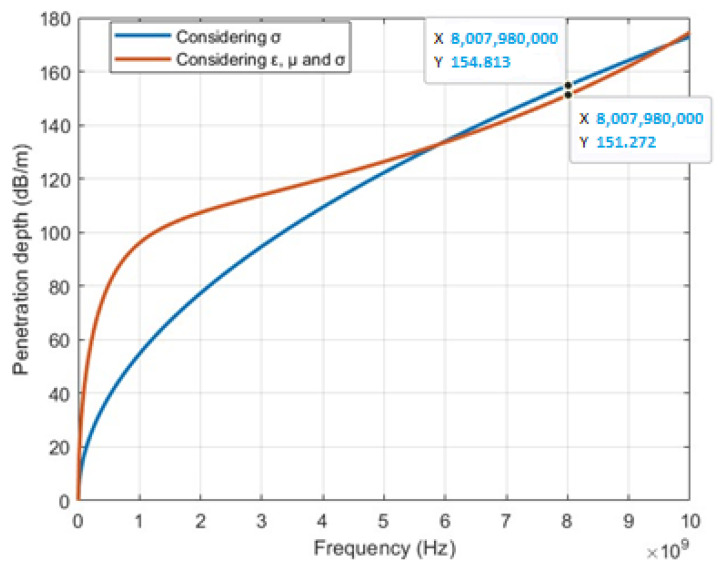
Depth of penetration.

**Figure 6 sensors-23-05480-f006:**
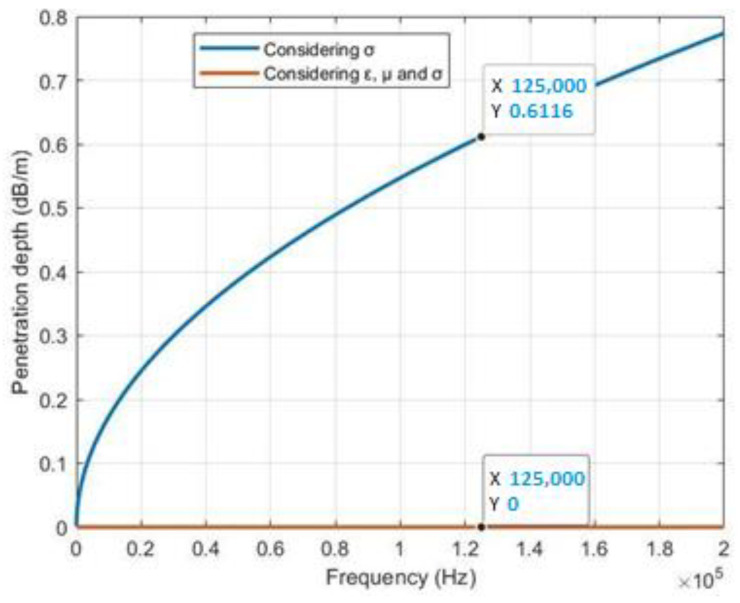
Graph of the depth of penetration in the order of KHz.

**Figure 7 sensors-23-05480-f007:**
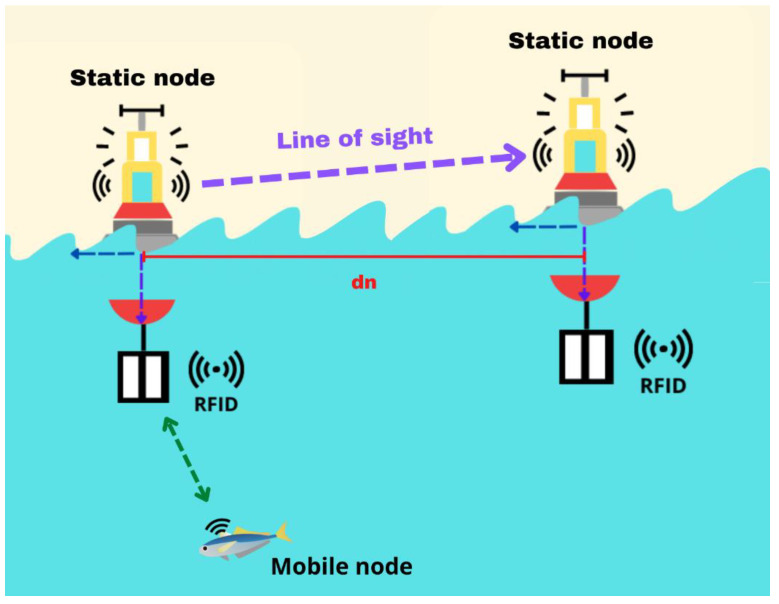
The first part of the second test scenario.

**Figure 8 sensors-23-05480-f008:**
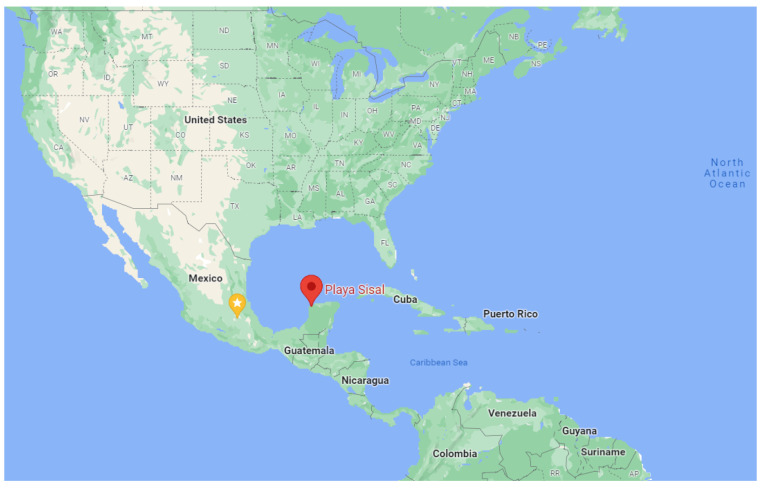
Location of Sisal Beach.

**Figure 9 sensors-23-05480-f009:**
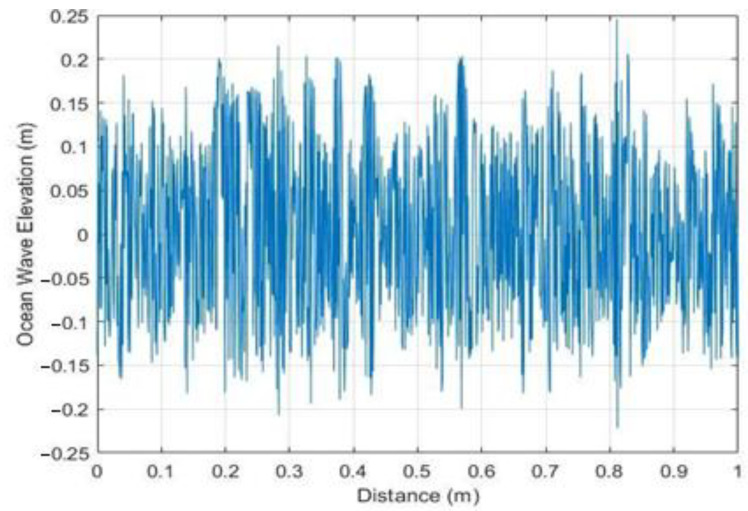
Waves at a distance of one meter.

**Figure 10 sensors-23-05480-f010:**
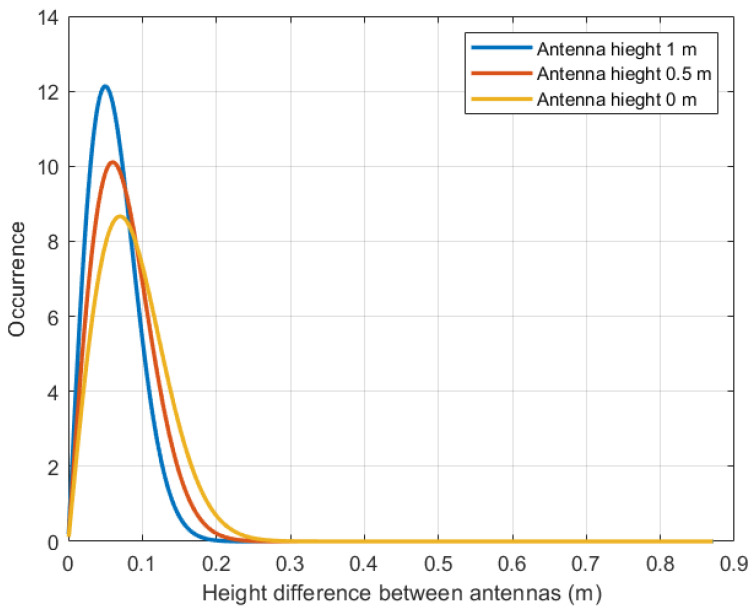
Probability distribution between static nodes: yellow line, 0 m; red line, 0.5 m; blue line, 1 m.

**Figure 11 sensors-23-05480-f011:**
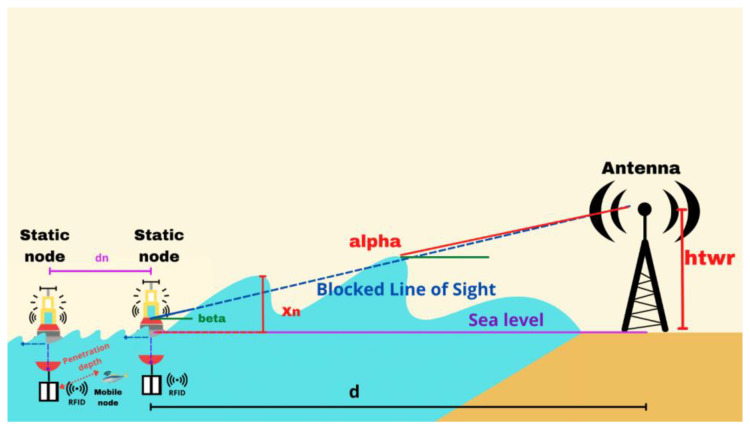
Representation of the second part of the second scenario.

**Figure 12 sensors-23-05480-f012:**
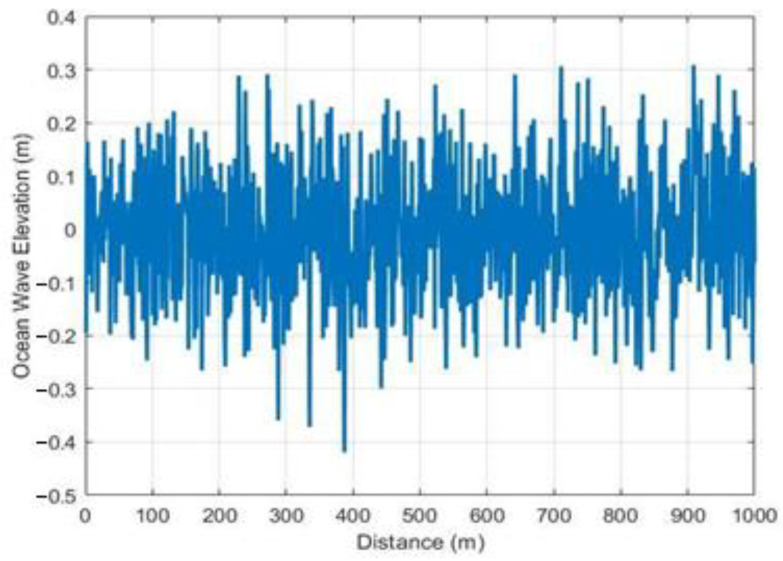
Wave elevation at a distance of 1 km.

**Figure 13 sensors-23-05480-f013:**
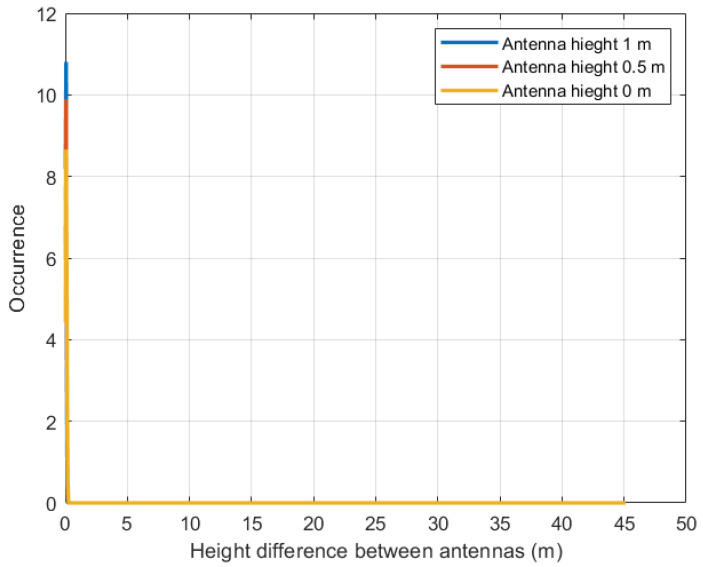
Probability distribution between the static node and antenna: yellow line, 0 m; red line, 0.5 m; blue line, 1 m.

**Table 1 sensors-23-05480-t001:** Database published by the Southeast Coastal Observatory [25].

ID	Significant_ Wave_Height	Peak_ Period	Mean_ Period	Measured_at	Lat	Updated_at
2802	0.8	34.12	18.7	20 March 2019 22:50:52 UTC	21.1646	20 March 2019 23:21:48 UTC
2803	0.25	10.24	11	20 March 2019 23:20:52 UTC	21.1646	20 March 2019 23:21:48 UTC
2806	0.39	25.6	13.36	20 March 2019 23:50:57 UTC	21.1646	21 March 2019 01:25:36 UTC
2807	0.62	34.12	15.08	21 March 2019 00:20:57 UTC	21.1646	21 March 2019 01:25:36 UTC
2804	0.71	25.6	13.56	21 March 2019 00:51:03 UTC	21.1645	21 March 2019 01:24:43 UTC
2805	0.77	20.48	14.96	21 March 2019 01:21:03 UTC	21.1646	21 March 2019 01:24:43 UTC
2808	0.65	25.6	14.68	21 March 2019 01:51:08 UTC	21.1646	21 March 2019 02:45:25 UTC
2809	0.6	34.12	15.16	21 March 2019 02:21:08 UTC	21.1646	21 March 2019 02:45:25 UTC
2810	0.98	25.6	17.08	21 March 2019 02:51:14 UTC	21.1646	21 March 2019 03:22:52 UTC
2811	0.85	34.12	17.54	21 March 2019 03:21:14 UTC	21.1646	21 March 2019 03:22:52 UTC
2812	0.66	25.6	14.68	21 March 2019 03:51:19 UTC	21.1645	21 March 2019 04:49:14 UTC
2813	0.48	25.6	17.12	21 March 2019 04:21:19 UTC	21.1646	21 March 2019 04:49:14 UTC
2814	0.71	34.12	18.92	21 March 2019 04:51:24 UTC	21.1646	21 March 2019 05:27:23 UTC
2815	0.65	34.12	16.14	21 March 2019 05:21:24 UTC	21.1646	21 March 2019 05:27:23 UTC
2816	0.52	25.6	13.84	21 March 2019 05:51:30 UTC	21.1646	21 March 2019 06:52:52 UTC
2817	0.52	10.24	11.66	21 March 2019 06:21:30 UTC	21.1646	21 March 2019 06:52:52 UTC
2818	0.64	10.24	10.44	21 March 2019 06:51:36 UTC	21.1646	21 March 2019 07:32:20 UTC
2819	0.71	10.24	12.72	21 March 2019 07:21:36 UTC	21.1646	21 March 2019 07:32:20 UTC
2820	0.78	34.12	15.8	21 March 2019 07:51:41 UTC	21.1646	21 March 2019 08:57:51 UTC
2821	1.17	34.12	19.56	21 March 2019 08:21:41 UTC	21.1646	21 March 2019 08:57:52 UTC

**Table 2 sensors-23-05480-t002:** Accuracy between static nodes.

Antenna Height of the Node (m)	Range of Difference between Antennas (m)	Probability (%)
0	0 to 0.2886	54.2
0.5	0 to 0.2573	70.6
1	0 to 0.2362	94.5

**Table 3 sensors-23-05480-t003:** Accuracy between the static node and terrestrial antenna.

Antenna Height of the Node (m)	Range of Difference between Antennas (m)	Probability (%)
0	0 to 0.3389	98.1
0.5	0 to 0.2968	99.2
1	0 to 0.2463	100

**Table 4 sensors-23-05480-t004:** Baseline.

Work	This Paper	[7]	[8]	[11]	[12]	[13]
Propagation method	MI	MI, acoustic, EM, optical	EM	MI	MI	MI
Transmission frequency (kHz)	125	No data	$1.9\times 10^{6}$	$134.5–13.56\times 10^{3}$	$125–13.56\times 10^{3}$	110
Penetration depth (dB/m)	0.6116	Only proposal	0.5 to 0.02	No data	0.0683	No data
Underwater data reception	YES	Only proposal	No data	YES	YES	YES
Sea surface data reception	YES	Only proposal	YES	NO	NO	NO
Public database wave sample	Sisal Beach Yucatan Mexico	No database	Controlled scenario database	No database	No database	No database

## Data Availability

Not applicable.

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
