# Peer review of "Analysis of Data Reception in the Communication Layer Applied to an Architecture of Mobile Sensor Networks in Marine Environments"

_sensors, 2023, doi:10.3390/s23125480_

Round 1
Reviewer 1 Report
The paper presents an important topic that is related to underwater wireless wireless technology. The following points should be taken into consideration:
First: Paper language, style and formatting
The paper needs a thorough review for the utilized style, grammar, and writing style. The following are some examples of the things that should be addressed:
1. Sometimes the acronyms are initial small and sometimes are initial capital. I suggest unifying that and use initial small throughout the paper. Further, the acronym should be defined once at the first place, for example: 36 RFID acronym is defined as initial small letters while in line 206 Significant Wave Height (SWH) is defined from initial capital words.
2. The paper writing needs improvement. For example: line 17:
“to check through the line of sight if the probability from the reception of data from the 17 antennas of the static nodes and from the terrestrial antenna.” (not complete sentence).
In Line 15, in the first one characterises ((not complete sentence).)
Line 150. And the cloud storage …. (you should start a sentence with And!!!)
In line 163: Characterization of data reception between the mobile node and the static node, is this a title or sub-title, then where is the proper formatting??? (e.g. 2.3 Characterization of data reception between the mobile node and the static node). The same sentence is repeated after Algorithm 1. Why?
Technical feedback
1. Line 133: so it does not intervene 133 in the natural behaviour of animals that works well in marine environments
What does it mean? can you elaborate more?
2. Fig needs more elaboration and description than the one provided on page 3. I did not understand what are these mobile objects until I saw Figure 4. I suggest describing Fig 1 in details on page 3
3. For the mobile nodes communication with the RFID, it is not clear. At the beginning you have mentioned it is EM induction, then on line 165, you said radio frequency waves. It is two different methods, can you elaborate in details how this communication is performed?
4. How about the communication range and distances between the mobile nodes and RFID nodes?
5. What is the paper contribution compared with the literature? I was hoping to see a table of comparison between this paper and what has been proposed in the literature highlighting the paper contribution.
The paper presents an important topic that is related to underwater wireless wireless technology. The following points should be taken into consideration:
First: Paper language, style and formatting
The paper needs a thorough review for the utilized style, grammar, and writing style. The following are some examples of the things that should be addressed:
1. Sometimes the acronyms are initial small and sometimes are initial capital. I suggest unifying that and use initial small throughout the paper. Further, the acronym should be defined once at the first place, for example: 36 RFID acronym is defined as initial small letters while in line 206 Significant Wave Height (SWH) is defined from initial capital words.
2. The paper writing needs improvement. For example: line 17:
“to check through the line of sight if the probability from the reception of data from the 17 antennas of the static nodes and from the terrestrial antenna.” (not complete sentence).
In Line 15, in the first one characterises ((not complete sentence).)
Line 150. And the cloud storage …. (you should start a sentence with And!!!)
In line 163: Characterization of data reception between the mobile node and the static node, is this a title or sub-title, then where is the proper formatting??? (e.g. 2.3 Characterization of data reception between the mobile node and the static node). The same sentence is repeated after Algorithm 1. Why?
Reviewer 2 Report
This paper discusses the analysis of data reception in a communication layer for a network of mobile and static nodes using radio frequency identification (RFID) technology at a frequency of 13,125 kHz. The objective is to implement this technology in marine environments, specifically for the Underwater Internet of Things (UIoT). The analysis is divided into two parts.
In the first part, the paper characterizes the penetration depth at different frequencies. It verifies that at a frequency of 125 kHz, there is data reception with a penetration depth of 0.6116 dB/m.
The second part simulates a sample of waves obtained from Sisal beach in Yucatan, Mexico. The simulation checks the line of sight to determine the probability of data reception from the antennas of the static nodes and the terrestrial antenna. The results show a maximum probability of 94.5% between the static nodes and a 100% probability of data reception between the static node and the terrestrial antenna when the antennas of the static nodes are positioned at an optimal height of 1 meter.
Based on these findings, the paper suggests that the proposed architecture can be implemented by adjusting the parameters mentioned in the paper and modifying the characteristics of the RFID system. This analysis opens the possibility of increasing the monitoring area using the proposed approach.
Strengths:
The article provides a comprehensive analysis of data reception in underwater environments, offering valuable insights. The obtained results, along with the proposed parameters, indicate that the proposed architecture can be successfully implemented in marine settings. Specifically, it is recommended to maintain a minimum antenna height of half a meter to ensure optimal data reception range. Moreover, the analysis suggests the possibility of adjusting certain parameters, such as the characteristics of the RFID systems, to enhance the penetration depth and facilitate greater separation between static nodes. This adjustment would ultimately result in a larger monitoring area for improved system performance.
Areas for Improvement
The predominant use of ultrasonic waves in the marine environment is well-established. However, the rationale behind analyzing radio waves in this context is to explore their potential applications and benefits. What is the rationale behind considering radio wave for this analysis?
To enhance the conclusion section, it is crucial to include future research directions and guidelines for reproducibility. It is recommended to provide insights into potential avenues for further investigation and suggestions for replicating the study. This allows researchers to build upon the findings and contribute to the advancement of knowledge in the field. By incorporating this information, the conclusion section becomes more comprehensive.
The literature review section appears to be lacking in comprehensiveness. It seems that recently published articles have been overlooked. It is advisable for the authors to include and highlight recent work to address the research gap more effectively.
Highlight the contribution in the bullet format which will attract the reader.
The analysis does not account for the dynamic characteristics of the environment, which may lead to different outcomes in real-world settings.
The clarity of the simulation environment needs improvement, and it is recommended to present the parameters in a table format. Additionally, the inclusion of an extensive analysis is necessary. Furthermore, comparing the results with the current state of the art is essential.
NC
Reviewer 3 Report
The authors should include problem statement(s) and objective(s) in the abstract section.
Overall, the introduction provides a good background and relevant references for the study. However, one possible improvement could be to provide a clearer and more specific statement of the research problem or objective early on in the introduction so that readers can better understand the motivation for the study.
Additional details and justification of some of the methods used, such as the selection of specific frequencies and the use of the Rayleigh distribution for probability calculations, are required to be discussed and justified in the research design section.
The authors should highlight and explain the modifications that they have made to their proposed method and align them with the problem or objectives of this research work.
The author should provide more information and justification on the experimental design and evaluation metrics used to analyze the data, which would also increase the rigor and reproducibility of the study.
The authors should compare their method with the benchmark solutions. This information can be included in the discussion section.
The results section presents the findings of the study in a clear and organized manner, with figures and tables used to illustrate the data. However, there are some areas where improvements can be made:
The section could benefit from more detailed explanations of the results and their significance. Some of the figures and tables are not explained thoroughly enough, which may make it difficult for readers to fully understand their meaning.
Figures and tables could be cited and incorporated into the text. Instead of simply stating that "Figure 5 shows the depth of penetration," the text could provide a more detailed explanation of the contents of the figure and what they represent. Figure 6 was included without being referred to in the text, and no additional information was provided. Figures 9 and 10 should be explained in greater detail because all of the subfigures are very similar and cannot be distinguished with the naked eye.
The authors might need to echo or highlight the modifications that they have made to their proposed method with the results that they obtained in solving the problem or achieving the objectives set in this research work.
The results appear to support the conclusions well. However, it might be helpful to include a discussion on the limitations of the study and potential areas for future research. Additionally, providing more information on the practical applications of the proposed architecture could help readers understand the potential impact of the study.
Reviewer 4 Report
This paper studies the data reception in the communication layer of mobile sensor networks in marine environments. First of all, I have to say that there are so many English Grammar and wording issues which significantly affect the reader’s understanding of the contents. Please try to improve it. I also have some comments and questions listed below.
Line 102-103: “permittivity” appears twice. Please fix it.
Table 1 (Algorithm 1): “??? penetration depth considering ?, ? y ?”, what does “y” mean?
“For the penetration depth considering ?,?0 y ?0”, no 0 for ? and ?.
Table 2 (Algorithm 2): “Regenerate a set of realization of ?? y ?? random”, what does “y” mean?
Line 251: “where said reader is submerged in water”
Line 277: “to verify if said reception exists”
Line 368: “when it said frequency only σ is …”
And many other places…
Is “said” a typo? Do you mean RFID?
Fig 9 is PDF and Fig 10 is CDF, is that correct? If yes, what’s the reason of presenting both?
Subfigure a, b, c of Fig 9 look almost the same. Similar in Fig 10. Are they correct? If you want to compare those subfigures and show their difference, it’s better to plot them in the same figure and probably zoom-in to some specific X-axis range to highlight the difference.
Fig 13: Could you provide a zoom-in version to the X-axis range of e.g. 0~1m?
There are so many English Grammar and wording issues which significantly affect the reader’s understanding of the contents. Please try to improve it.
Round 2
Reviewer 2 Report
The authors have addressed all the concerns. Thus I do not have any new comments. Minor issue: Add recently published articles in the literature review.
NC
Reviewer 3 Report
The authors only addressed the comment partially. The following comments have not been addressed yet:
1. Overall, the introduction provides a good background and relevant references for the study. However, one possible improvement could be to provide a clearer and more specific statement of the research problem or objective early on in the introduction so that readers can better understand the motivation for the study.
2. The authors should highlight and explain the modifications that they have made to their proposed method and align them with the problem or objectives of this research work.
3.The authors should compare their method with the benchmark solutions. This information can be included in the discussion section.
4. It might be helpful to include a discussion on the limitations of the study and potential areas for future research
5. Lines 359 and 365 should be in Figure 9 instead of Figure 10.
6. I'm not sure what the intention is behind including Figure 13. There is limited text explaining this figure. Are Figure 13 and Table 5 representing the same information?
7. The y-axis of Figure 13 is labeled "probability". The value shown is in %, or what? Why is this value different from Table 5?
